# InfoAug: Mutual Information Informed Augmentation for Representation Learning

## Abstract

Representation learning methods utilizing the InfoNCE loss have demonstrated considerable capacity in reducing human annotation effort by training invariant neural feature extractors. Although different variants of the training objective adhere to the information maximization principle between the data and learned features, data selection and augmentation still rely on human hypotheses or engineering, which may be suboptimal. For instance, data augmentation in contrastive learning primarily focuses on color jittering, aiming to emulate real-world illumination changes. In this work, we investigate the potential of selecting training data based on their mutual information computed from real-world distributions, which, in principle, should endow the learned features with better generalization when applied in open environments. Specifically, we consider patches attached to scenes that exhibit high mutual information under natural perturbations, such as color changes and motion, as positive samples for learning with contrastive loss. We evaluate the proposed mutual-information-informed data augmentation method on several benchmarks across multiple state-of-the-art representation learning frameworks, demonstrating its effectiveness and establishing it as a promising direction for future research. The data and code will be available for further investigation.

## 1 Introduction

Self-supervised learning has witnessed remarkable advancements in various domains in recent years, including contrastive self-supervised learning(Oord et al., 2018; Ye et al., 2019), generative self-supervised learning(Kingma & Welling, 2013; Goodfellow et al., 2020; He et al., 2021) and so on. These approaches leverage different proxy tasks to model unlabeled data, often surpassing their supervised counterparts in performance. Among these methods, substantial efforts have been dedicated to the exploration of contrastive learning paradigms, yielding impressive achievements exemplified by influential models such as SimCLR (Chen et al., 2020) and MoCo (He et al., 2020).

The fundamental essence of contrastive learning is instance discrimination(Wu et al., 2018). Specifically, during the training process, contrastive learning model aims to bring different augmented views of the same entity closer in the representation space, while pushing apart different entities. Admittedly, such way of positive sample selection enables good representation learning by encouraging the model to be view invariant.

However, upon recalling the human visual learning process, we realize that human's way to determine positive samples extends beyond instance patches discrimination. What are positive samples? By its nature, positive samples could only come either "from same entity" or "cross different entities". The "from same entity" part is well implemented by traditional contrastive learning via view-based data augmentation, while the "cross different entities" part is not yet addressed. Based on this observation, we argue that utilizing mutual information(Shannon, 1948) is one of the most natural measures to discover cross entities positive pairs, which share high mutual information content. Imagine a scenario where two birds flying together in the sky and a toy bird randomly walking on the ground. Although all of them share a same look, we could still correctly discriminate two birds flying as positive samples, for knowing the position of one real bird reduces the uncertainty of another, while toy bird won't tell us with anything about those two real birds.

Driven by this motivation, we propose InfoAug, a mutual information informed data augmentation technique towards a unified positive sample determination for contrastive learning. Specifically,

for each patch in the scene, we examine its mutual information with all other patches appearing in the same scene, and the one that exhibits the highest mutual information are considered to be its cross-patch positive samples, denoted as each other's twin patch. By asking where do positive samples really exist, InfoAug combines the traditional view-level **same entity** data augmentation and our novel **cross entities** mutual information informed data augmentation together, which is a more unified data augmentation approach for contrastive learning.

For demonstration of our method, we gather certain standard video dataset where we only focus on the patches in the first frame of each video for learning, the left frames are only used to estimate mutual information between patches attached to the first frame. We also won't tap into any temporal contrastive learning field(Pathak et al., 2017; Wang & Gupta, 2015) so as to control a single variable to demonstrate its effectiveness. However, we will illustrate how temporal contrastive learning could be combined towards a ultimately unified contrastive paradigm in the future work section, it is just not the focus of this work. In short, we split the first frame of each video to several patches, and do patch-level tracking to obtain their motion trajectories, which we will use to empirically estimate the mutual information between those patches. The patch that shares highest mutual information with a given patch, which we denoted as its twin patch, will be treated as its positive pairs while training. Such method encourage the model to be mutual information aware, which proves to consistently improve the model capacity.

We will demonstrate the effectiveness of this method through comparisons with seven prominent baselines in the following sections. We evaluated its performance on downstream classification tasks using CIFAR-10((Krizhevsky et al., 2009), CIFAR-100 (Krizhevsky et al., 2009), and STL-10 (Coates et al., 2011). The results consistently surpass the original baselines, showcasing varying degrees of improvement. In summary, our contributions are as follows: 1. We propose a novel method for determining positive samples that better aligns with real-world models and human visual learning. 2. We illustrate a very simple yet effective "dual-projection-branch contrastive learning" pipeline to accommodate our proposed MI-guided positive pair selection.

## 2 RELATED WORK AND PRELIMINARY

### 2.1 MUTUAL INFORMATION

**Mutual information as a correlation measure** The mutual information $\mathcal{I}$ is normally defined as the joint distribution of two random variables $X$ and $Y$:

$$\mathcal{I}(X;Y) = \sum_{y \in \mathcal{Y}} \sum_{x \in \mathcal{X}} p(x,y) \log \left( \frac{p(x,y)}{p(x)p(y)} \right)$$

where $p(x,y)$ is the joint probability distribution, and $p(x)$, $p(y)$ are marginal probability distributions. A more intuitive form could be given as:

$$\mathcal{I}(X;Y) = H(X) - H(X|Y) = H(Y) - H(Y|X)$$

where $H()$ represents entropy, a measure for uncertainty. Compared with linear dependency given by Covariance/Correlation, mutual information shows a more general form of dependency by measuring conditional uncertainty reduction. Thus, leveraging mutual information serves as a more appropriate measure for determining relevancy between objects described by random variables, and further discovering positive samples in contrastive learning.

**Empirical Estimation of Mutual Information** Estimating mutual information is difficult due to its association with non-linearity. When we face small-sample-size, high-dimensionality, unbalanced-sample, this often brings estimation bias to the result (Kraskov et al., 2004; Paninski, 2003). There are currently three major lines of work with mutual information estimation, namely parametric, non-parametric and neural estimation paradigm (Walters-Williams & Li, 2009). Their respective representatives include Maximum-likelihood (Suzuki et al., 2009), K-nearest-neighbour (Kraskov et al., 2004) and MINE (Mutual-information-neural-estimator) (Belghazi et al., 2018).

In this paper, we choose "3KL" based on K-nearest-neighbour principle proposed by Kraskov et al. (2004) to estimate mutual information. It is widely known as a good estimator as it strikes a good balance between inference accuracy and speed with limited sample size. The core idea is to utilize

K-th distance between samples to approximate original density of the joint distribution so as to calculate mutual information. A mathematical form of it is as follows:

$$\widehat{\mathcal{I}_{3KL}}(x;y) = \psi(n) - \psi(k) + \frac{1}{n}\sum_{i=1}^{n}\log\left(\frac{\epsilon_X^k(x_i) \cdot \epsilon_Y^k(y_i)}{\left(\epsilon_P^k(p_i)\right)^2}\right) \qquad (1)$$

where $\epsilon_X^k(x_i)$, $\epsilon_X^k(y_i)$, $\epsilon_X^k(p_i)$ refer to the k-nearest distance of the i-th sample of $x$, $y$ and their joint variable denoted as $\hat{P}$, and $\psi()$ refers to the Di-gamma function. In our method, we adopt this approach for mutual information estimation between two random variable describing the position of two patches of interest.

## 2.2 CONTRASTIVE LEARNING

The essence of contrastive learning lies in instance discrimination, specifically, it aims to bring variant of a same entity closer and push different entities farther away in embedding space. Recently, most works in contrastive learning concentrated on exploiting more different views of an image or a sequence of images. MoCo (He et al., 2020) leveraged a momentum updated queue to replace costly memory bank, while maintaining diversity of negative pairs. SimCLR (Chen et al., 2020) focused on composition of multiple data augmentation operations, which proved to be crucial in discovering more different views of images. Different from previous works, BYOL (Grill et al., 2020) demonstrated that with careful design of the framework, positive pairs are enough for contrastive learning, without participant of negative pairs.

Going beyond 2D images, some works utilize video sequence to bring more different views of a same scene or object Dave et al. (2022); Pan et al. (2021). Taking advantages of both spatial and temporal information of video sequence, Qian et al. (2021) managed to boost the performance of video representation. Other works mainly focus on learning from future prediction and sorting Jing et al. (2019); Kim et al. (2018); Fernando et al. (2017). Wang et al. (2019) extracted visual features from the prediction of both motion and appearance statistics along spatial and temporal dimensions. And Xu et al. (2019) learned the spatiotemporal representation of the video by predicting the order of shuffled clips from the video.

Utilizing mutual information in describing the learning objective has also attracted people's eyes recent years(Torkkola, 2003; Tschannen et al., 2019; Hjelm et al., 2018). Wu et al. (2020) illustrated that a family of algorithms were maximization of a lower bound on the mutual information between two or more "views" of an image and proposed their technique which generalized the InfoNCE objective (Gutmann & Hyvärinen, 2010; Poole et al., 2019). Bachman et al. (2019) proposed a method to extract multiple views of a local spatial-temporal context by maximizing mutual information. Klein & Nabi (2023) utilized a mutual information-based contrastive learning framework to learn sentence embedding which enforced the structural consistency across augmented views for every sentence.

Our work also take video sequence as input for mutual information estimation and further twin patch discovery. However, we stay within the non-temporal contrastive learning area since our aim is to propose an unified data augmentation technique for positive sample determination, and this will be illustrated later. The frames after the first frame are merely for our twin patch selection technique InfoAug, and they are not involved in training.

## 3 METHOD

**Overview.** In this section, we aim to illustrate the principle that drives InfoAug and detailed technique. We will first make necessary notations to facilitate further description in 3.1. Then, at the core of our idea, we'll illustrate how to utilize mutual information to guide twin patch discovery in 3.2. Finally, we will show our two-branch training pipeline, which accommodates our InfoAug towards a more unified contrastive learning paradigm.

## 3.1 NOTATIONS

Given a video $\mathbf{S}_k$ sampled from the dataset $\{\mathbf{S}_k\}_{k=0}^{K}$, Notice that we focus on the first frame $\mathbf{I}_k$, while the subsequent frames are used to estimate mutual information only. Then, we split $\mathbf{I}_k$ into $N$

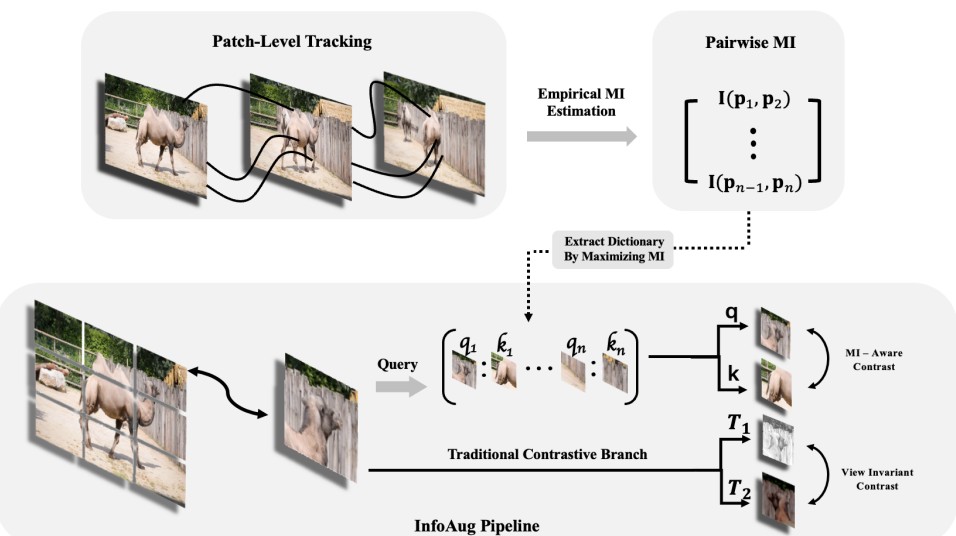

Figure 1: This figure shows the general pipeline of **InfoAug**. Through video tracking, we estimate the pairwise mutual information between patches attached to a same scene, which we base on to construct a "twin patch" dictionary. While training, besides traditional data augmentation($T1, T2$), we use twin patch dictionary to form another positive pair, thereby injecting mutual information awareness into the model, towards a more unifed data augmentation paradigm.

patches with equal size, denoted as $\{\mathbf{P}_{k,i}\}_{i=0}^{N}$, where $i$ is the index of the patch in the first frame of a video. These patches are fundamental elements(data unit) in our experiments, that is, all positive pairs refers to relationship between two patches.

## 3.2 MUTUAL INFORMATION GUIDED TWIN PATCH SELECTION

The principle of our work is similar to the notion of wave function in physics: **a sequence of observation of an object's position is a collection of samples from their own distribution**(Aharonov et al., 1993; Tsutsumi, 1987). That is, by performing patch-level tracking, we are essentially sampling from the real-world distribution of an object covered by that patch. Taking a step forward, if we simultaneously track two arbitrary patches at the same time, the collection of observations are undoubtedly drawn from their joint distribution, with which you could make empirical inference on any statistical measure, including their mutual information. Based on this idea, we will show in the following subsection: 1. how do we obtain twin patch all the way from patch-level tracking and 2. how our alignment techniques help to improve the practicability.

### 3.2.1 FROM TRACKING TO TWIN PATCH SELECTION

As shown in Fig 2(a), traditional contrastive learning mainly focuses on different augmented views of a same image patch. Whereas in our system, we will not only treat the aforementioned different views of an image patch as positive samples, but also the patch itself and its twin patch, with whom it exhibits the highest mutual information. Based on this principle, we dubbed our novel data augmentation technique **InfoAug**, which serves as an unified way to discover positive samples utilizing mutual information. To facilitate further notation, we name the patch that share the highest mutual information content with patch of interest, its **twin patch**.

**Algorithm.** Given a video $\mathbf{S}_k$, we first slice the first frame, $\mathbf{I}_k$, evenly to $N$ patches. As shown in Fig 1, we assign a representative point in the center of each patch, denoted as $\mathbf{p}_{k,i}$. Then, we adopt an off-the-shelf tracking model like TAPIR (Doersch et al., 2023) to track all representative points within $\mathbf{I}_k$ along the whole sequence and get their trajectories $\{Traj_{k,i}\}$. To align 2-D trajectories in camera reference frame with real world 3-D position, we consider to incorporate depth information to form the 3-D trajectory. Here, we use MiDaS (Ranftl et al., 2020) to generate depth information for all frames along the whole sequence. The depth value and 2-D trajectories will be concatenated to scale $\{Traj_{k,i}\}$ up to 3-D trajectories. After that, we could empirically estimate mutual information

$\mathcal{I}(\mathbf{P}_{k,i_0}, \mathbf{P}_{k,i_1})$ between any two patches $i$ and $j$ as follows:

$$\mathcal{I}(\mathbf{P}_{k,i}, \mathbf{P}_{k,j}) = \widehat{\mathcal{I}_{3KL}}(Traj_{k,i}, Traj_{k,j}), \quad \text{where } i, j \in \{0, 1, 2, \ldots, N\} \tag{2}$$

Thus, for any patch, $\mathbf{P}_{k,i}$, in a first frame of a given video, we choose its twin patch to be patch of index $j$ who shares highest mutual information with $\mathbf{P}_{k,i}$ as follows:

$$j = \arg\max_{j \neq i} \mathcal{I}(\mathbf{P}_{k,i}, \mathbf{P}_{k,j}), \quad \text{where } j \in \{1, 2, 3, \ldots, N\} \tag{3}$$

By looping through all videos with the above mentioned algorithm, we obtain a twin patch for all patches in all videos. To be specific, we'd like to say that we essentially hold a "twin patch dictionary" where the keys are all patches in the dataset and the values are their corresponding twin patch who empirically showed to share highest mutual information with them.

### 3.2.2 ALIGNMENT WITH REAL-WORLD MODEL.

In this subsection we share an important engineering technique called "alignment with real-world model". As its name suggests, we are motivated by the fact that there exists deviation between real world position distribution and position samples captured by camera. Consequently, it may lead to unreasonable twin patch selection. To this end, we will analyze possible reasons of deviation and share our way to cope with it.

**Not enough entropy exhibited.** A video containing $F$ frames provides $F$ observations for empirical mutual information estimation between patches. However, such a small amount of observation may not be enough for us to correctly determine positive sample for those patches who don't exhibit enough entropy. For example, we can't determine whether two objects are positive samples or not if they seldom make any movement in real-world, though they may share high mutual information. In real world, this is not a problem since we have access to observe the world with a very long time span. However, in a short video, such purification is necessary for reasonable estimation.

To this end, we filter out those points with small entropy and only consider points that exhibit high entropy. There are multiple ways to do the split, and we conform to "Maximum Gap Algorithm" (Hoberman et al., 2005). Using this algorithm, we only focus on points with high entropy, to avoid unreasonable positive patches. We will also show in 4, that incorrectly selected positive sample will do harm to contrastive learning.

**The motion of camera.** From above, we know that we should focus on points with actual high entropy. However, in some videos, the camera may move together with the moving object, which makes the moving object appear to be static, for instance, a horse racing game recording. This will reverse the actual "high-entropy" points with "low-entropy" points since the reference frame is no longer earth-reference-frame.

To tackle this problem, we first determine whether the camera is in motion by comparing the entropy of patches on the outermost ring of the image with the entropy of other internal patches. If the one on outermost ring is bigger, then it is most likely that the camera is in motion, then we would select points with low estimated entropy to perform "twin-patch" selection, and vice-versa.

### 3.3 TWO BRANCH TRAINING

**Motivation.** For a given patch, assigning its twin patch as its positive sample encourages the model to be "mutual information aware", while the traditional data augmentation encourages the model to be "view invariant". To simultaneously accommodate these two objective, we use the "two branch learning" formulation as illustrated in Fig 2.

**Model and Loss.** As shown in Fig2, our pipeline consists of two main branch: one for traditional same-patch-different-view contrastive learning, and another for our twin-patch contrastive learning. Given a batch of patches $p = \{\mathbf{P}_{k_0,i_0}, \mathbf{P}_{k_1,i_1}, \ldots, \mathbf{P}_{k_b,i_b}\}$ and its twin patch $p^{(twin)} = \{\mathbf{P}_{k_0,i_0}^{(twin)}, \mathbf{P}_{k_1,i_1}^{(twin)}, \ldots, \mathbf{P}_{k_b,i_b}^{(twin)}\}$, we will first augment $p$ by two different transformations $T_1$ and $T_2$ for the first branch propagation, and get $p^{(1)} = \{\mathbf{P}_{k_0,i_0}^{(1)}, \mathbf{P}_{k_1,i_1}^{(1)}, \ldots, \mathbf{P}_{k_b,i_b}^{(1)}\}$ and $p^{(2)} = \{\mathbf{P}_{k_0,i_0}^{(2)}, \mathbf{P}_{k_1,i_1}^{(2)}, \ldots, \mathbf{P}_{k_b,i_b}^{(2)}\}$. Noted that the transformation $T_1$ and $T_2$ conform to that of the original baseline framework, for example, it includes $Random Resized Crop$, $Random Vertical Flip$,

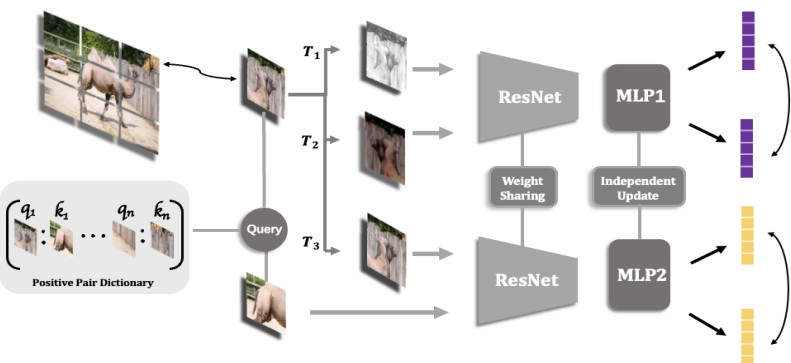

Figure 2: This figure shows a two-branch learning formulation adopted by **InfoAug**, where the backbone use weight-sharing while the **two projection heads** is independently updated, which aims for a decoupling of the two learning objectives: 1.view-invariant and 2.mutual information aware.

$ColorJittering$ and so on if we are comparing with SimCLR. Then, $p$, $p^{(1)}$, $p^{(2)}$, and $p^{twin}$ will all go into a same encoder $f()$, encoded as $\mathbf{h}^{(1)} = f(\mathbf{p}^{(1)})$, $\mathbf{h}^{(2)} = f(\mathbf{p}^{(2)})$, $\mathbf{h} = f(\mathbf{p})$, $\mathbf{h}^{twin} = f(\mathbf{p}^{twin})$. After that, $\mathbf{h}^{(1)}$ and $\mathbf{h}^{(2)}$ will go through projection head 1, which is responsible for "view invariant" embedding, and $\mathbf{h}$ and $\mathbf{h}^{(twin)}$ will go through projection head 2, which is responsible for "mutual information aware" cross patch embedding. Ultimately, we get $\mathbf{z}^1$, $\mathbf{z}^2$, $\mathbf{z}$ and $\mathbf{z}^{twin}$.

After that, we calculate the loss by a weighted average of the original contrastive learning adopted, respectively on $(\mathbf{z}^1, \mathbf{z}^2)$, $(\mathbf{z}, \mathbf{z}^{twin})$:

$$\mathcal{L} = \mathcal{L}(\mathbf{z}^1, \mathbf{z}^2) + \lambda * \mathcal{L}(\mathbf{z}, \mathbf{z}^{twin}) \tag{4}$$

where $\mathcal{L}$ is original contrastive loss(for example, it is NTXentLoss() for SimCLR and NegativeCosineSimilarity() for BYOL), $\lambda$ is a weighted factor to adjust the balance between the two objectives.

This two branch formulation helps to decouple the two learning objectives into two projection heads that do not share weight, allowing them to be better tailored for their own task without being influenced by other learning objective.

### 3.4 Implementation Details

**Data.** We use DAVIS2018 (Caelles et al., 2018) and GMOT40 (Bai et al., 2021) to build our pre-training dataset, containing $\mathbf{K} = 90$ videos. And we use $N = 40$ patches for each frame in the sequence. We use a batch size of 100 for training. One possible concern is that why the pre-training is done in such a relatively small video dataset, we have performed experiment on larger dataset and we will share detailed analysis and potential future work to this problem in 5.1.

**Model.** We use the standard ResNet-18 (He et al., 2015) as the backbone encoder and we followed standard implementation in lightly for each baseline's model neck and model head. Here must not induce any bias, since for all baseline frameworks, our method(its InfoAug counterpart) use exactly the same architecture for model training.

**Training**. We utilize lightly framework(Susmelj et al., 2020) which serves a wrapper for standard pytorch code for implementations of all the baselines. We followed the baseline's standard settings in optimizer,scheduler and data augmentation, fixing the starting learning rate to be 0.02 across all the models.

## 4 Experiments

In this section, we will evaluate the effectiveness of InfoAug. First we will show the main comparison between all well-known baselines and their InfoAug-counterparts on image classification task on multiple datasets in 4.1. Then, we will further evaluate the effectiveness of mutual information-informed augmentation in 4.2. Next, we will illustrate the difference in performance with/without

dual-branch formulation in 4.3. Lastly, we will give ablation studies on some key hyper-parameters for further comprehension in 4.4.

**Experiment settings** We evaluate the capability of the backbone encoder trained with seven well-known SOTA frameworks: SimCLR, BYOL, SimSiam (Chen & He, 2020), MoCo, NNCLR (Dwibedi et al., 2021), VICReg (Bardes et al., 2022) and TiCo (Zhu et al., 2022) and their InfoAug-counterparts, on three datasets: CIFAR-10, CIFAR-100, and STL-100, for image classification task. We will not evaluate it on larger dataset like ImageNet, because our pre-training dataset is relatively small due to some practical reason with in-the-wild video dataset (will be discussed in 5). We extract the pre-trained backbone and add it with a three-layer classification head to do linear probing on the training split of these dataset and evaluate on their test split, we will show detailed results in 4.1.

## 4.1 MAIN COMPARISON

In this section, we will show the performance of seven state-of-the-art frameworks on downstream classification task and their InfoAug-counterparts with standard settings($\lambda = 1$; all models trained with 200 epochs). We will further perform ablation studies on these hyper-parameters later.

Table 1: Evaluation on image classification with linear probing on CIFAR-10, STL-10 and CIFAR-100, between seven strong baseline frameworks and its InfoAug counterparts.

| | CIFAR-10 | | STL-10 | | CIFAR-100 | |
|---|---|---|---|---|---|---|
| Frameworks | Original | with InfoAug | Original | with InfoAug | Original | with InfoAug |
| SimCLR | 66.44 | **67.48** | 58.38 | **60.18** | 37.02 | **37.10** |
| BYOL | 60.52 | **61.88** | 54.16 | **54.83** | 30.31 | **32.40** |
| SimSiam | 54.22 | **56.69** | 48.46 | **51.37** | 24.38 | **26.03** |
| MoCo | 61.67 | **62.24** | 53.25 | **54.33** | 30.78 | **31.40** |
| NNCLR | 62.97 | **63.57** | 57.07 | **57.12** | 32.31 | **32.97** |
| VICReg | 68.87 | **70.03** | 60.66 | **60.77** | 40.51 | **42.70** |
| TiCo | 62.14 | **63.35** | 53.63 | **57.36** | 33.47 | **35.37** |

The result demonstrates a consistent improvement on the encoder's capability since it shows a boost in performance for every baseline-benchmark combination. The results well proves that InfoAug is a framework-invariant technique that applies to any sort of training pipeline as long as it follows a general contrastive paradigm. Indeed, the improvement varies across different frameworks, which shall be reasonable since different methods differs in their way of performing data augmentations and also calculating loss based on the joint-embeddings, which all influence their compatibility with InfoAug.

## 4.2 EFFECTIVENESS OF MUTUAL INFORMATION BASED SELECTION

To demonstrate the effectiveness of mutual information informed postive sample selection, it is necessary to compare the mutual-information selecting algorithm for determining "twin patch", with randomly selecting patch as "twin patch" to train the encoder. Indeed, the boost in performance may merely come from the fact that InfoAug get access to additional patches from the same image when optimizing the model. We argue that although such observation(additional patch from same scene raises performance), if true, is also of value, our aim is to rigorously prove the effectiveness of InfoAug so as to drive attention to this natural relationship between mutual information and contrastive learning.

Indeed, access to additional patch from same scene sometimes helps with model performance but the results show that it is rather a random perturbation than a consistent improvement. The experiment demonstrates that the effectiveness of mutual information informed data augmentation technique, which supports the principle that mutual information should be examined as the key factor for positive sample determination.

Table 2: Evaluation on image classification with linear probing on CIFAR-10, STL-10 and CIFAR-100, between seven strong baselines and our pipeline formulation, but with randomly selected "twin-patch"("Random" for random twin patch), i.e. no mutual information measure involved.

| Frameworks | CIFAR-10 | | STL-10 | | CIFAR-100 | |
|---|---|---|---|---|---|---|
| | Original | Random | Original | Random | Original | Random |
| SimCLR | 66.44 | **67.40** | **58.38** | 58.18 | **37.02** | 36.69 |
| BYOL | 60.52 | **61.12** | 54.16 | **56.66** | 30.31 | **32.27** |
| SimSiam | 54.22 | **55.44** | 48.46 | **50.22** | 24.38 | **26.53** |
| MoCo | **61.67** | 60.30 | **53.25** | 53.02 | 30.78 | **30.68** |
| NNCLR | 62.97 | **63.04** | 57.07 | **57.88** | **32.31** | 32.99 |
| VICReg | **68.87** | 68.55 | **60.66** | 59.33 | **40.51** | 40.28 |
| TiCo | **62.14** | 62.09 | 53.63 | **56.11** | 33.47 | **34.27** |

## 4.3 COMPATIBILITY OF DUAL-BRANCH FORMULATION WITH VARIOUS FRAMEWORKS

Although dual-branch formulation heuristically fits with InfoAug since it both perform "view-invariant" and "mutual information aware" encoding, we here test the compatibility of this formulation with different frameworks to facilitate further research.

Table 3: Evaluation on image classification with linear probing on CIFAR-10, STL-10 and CIFAR-100, between InfoAug without dual-branch formulation and InfoAug with dual-branch formulation.

| Frameworks | CIFAR-10 | | STL-10 | | CIFAR-100 | |
|---|---|---|---|---|---|---|
| | Single | Dual | Single | Dual | Single | Dual |
| SimCLR | 67.17 | **67.48** | 59.92 | **60.18** | 37.08 | **37.10** |
| BYOL | 61.55 | **61.88** | **55.86** | 54.83 | 32.33 | **32.40** |
| SimSiam | 56.67 | **56.69** | 51.11 | **51.37** | **27.62** | 26.03 |
| MoCo | **62.61** | 62.24 | **55.25** | 54.33 | **33.16** | 31.40 |
| NNCLR | **63.86** | 63.57 | 57.06 | **57.12** | 32.87 | **32.97** |
| VICReg | 69.09 | **70.03** | 60.03 | **60.77** | 41.29 | **42.70** |
| TiCo | 63.01 | **63.35** | 56.00 | **57.36** | 33.34 | **35.37** |

From the experiment results, most frameworks do benefit from the dual-branch formulation. An exception here is MoCo, who turns out to always perform better with single branch InfoAug pipeline. We argue it might be the reason that the effect of traditional projection head and InfoAug projection head become unified when faced with a large dynamic dictionary. In summary, decoupling the two learning objective into two projection head helps with model learning, but is up to users' choice for further application

## 4.4 ABLATION STUDY

**Ablation on λ.** We visualize the model performance across the seven baseline framework on CIFAR-10 when the weighted factor are set to $[0, 0.5, 1.0, 1.5, 2.0]$ in Fig 3.

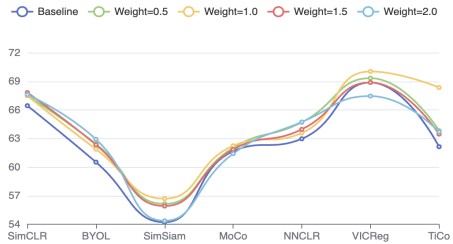

Figure 3: Ablation study on the weighted factor

Each line represent the performance related to a single weighted factor. We could see that the model perform the worst when $\lambda$ is set to 0, which correspond to the baseline method without InfoAug. We also learn from the ablation that setting the value too high can also harm the model. We argue that this is also reasonable due to the same reason as when $\lambda = 0$, which break the balance between the two above mentioned learning objective. The point is further proved by the fact that the model yields the best performance when $\lambda$ is set to 1, which strikes a good balance for model to simultaneously be view invariant and mutual information aware.

We also performed ablation study on training epoch and different size of training samples to demonstrate the effectiveness of the method. For the limitation of space in main text, we conserve the illustration of these studies to A.

## 5 DISCUSSION

### 5.1 CURRENT LIMITATIONS ON IN-THE-WILD LARGE DATASET

**Insufficient observation.** For some in-the-wild large scale video datasets like TrackingNet(Muller et al., 2018), GOT10K(Huang et al., 2019) and TAP-kinectics(Doersch et al., 2023), the number of frames in a video sequence varies a lot from video to video. This may result in insufficient information for mutual information estimation and the determination of appropriate "twin patch", due to nearly zero observed entropy or biased observation within too short period.

**Camera jittering.** Short term camera jittering may reduce the discrepancy between real-world position and estimated position in camera reference frame. This will inevitably lead to false positive sample with direct mutual information selection, which we argue is what offsets the advantage of InfoAug in the evaluation on those large scale datasets.

### 5.2 FUTURE WORKS

**More points to represent a patch.** A plausible way to mitigate the bias caused by insufficient observation and camera jittering could be assigning more points to a patch and using the concatenation of all points positions as the overall random variable for mutual information calculation. However, one problem of this solution lies in the current "3KL" estimation method(Kraskov et al., 2004), which may not be suffice for the increase of dimension. It requires us to turn to method like MINE(Belghazi et al., 2018), which will definitely take more time to finish.

**Incorporate temporal information towards a unified contrastive learning paradigm.** The path towards an ultimately unified approach to contrastive learning would be to combine our method with temporal contrastive learningPathak et al., 2017. Here, our work shows how to determine positive samples within a time frame, that is, without time dimension, while Wang & Gupta (2015) proposed a method on how to determine positive samples across the time dimension by tracking the objects. These two methods could be naturally combined with only one tracking and make fully use of the whole video sequence. We argue that it is the key step towards a generally unified contrastive learning paradigm.

## 6 CONCLUSION

In this paper, we investigate the problem of positive sample selection which is the key of self-supervised contrastive learning. We notice that traditional contrastive learning only answers the "within entity" part of the question but haven't deal with the "cross entity" part of it. Leveraging the natural bond between Mutual Information and cross entity positive samples, we proposed InfoAug, an more unified method which combines traditional view-based augmentation and our mutual-information-aware cross patch augmentation, based on empirical cross patch mutual information estimation. We test our method on extensive state-of-the-art baseline framework on multiple datasets, the result shows a consistent improvement on every baseline-benchmark combination, yielding various degree of progress.

We believe that InfoAug provides a playground for future work to explore on with mutual information based contrastive augmentation. More concretely, InfoAug's work on a more unified method on spatial level could be naturally fused with existing temporal contrastive method, which we believe is important towards a ultimately unified scene understanding framework.

**Ethics Statement:** We have read the ICLR Code of Ethics and ensures that this work follows it. All data and pre-trained models used in our experiments are publically available and has no ethical concerns.

**Reproducibility Statement:** To help readers reproduce our experiments, we provided detailed preliminary of our mutual information estimation algorithm in Section A.1, a breakdown of our proposed method in Section A.2, and extensive ablation study on related hyper-parameter in Section A.3. Since our work proposes an novel method working in slightly different experiment settings, we also provide all relative details in Section 3.4. Lastly, We also plan to release the source codes to ensure the reproducibility of this paper.

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

# A APPENDIX

## A.1 MORE ON "3KL" MUTUAL INFORMATION ESTIMATION ALGORITHM

In this section we give a more detailed explanation on the principle behind the method "3KL" that we deployed to estimate mutual information. Also, we illustrate how to structure the input to before feeding it the "3KL".

Kraskov et al. (2004) proposed a novel approach to estimate (joint) entropies using nearest neighbors in a given dataset. Instead of directly estimating the density of the distribution, they leveraged the distance to the k-th nearest neighbor as an approximation, as shown in Fig 5. A more intuitive explanation is, the density should be higher for the area where the k-nearest neighbour is closer.

By separately estimating the three entropies, namely the entropy of $\mathbf{X}$, denoted as $H(X)$, and the entropy of $\mathbf{Y}$, denoted as $H(Y)$ and the joint entropy, denoted as $H(X;Y)$, the method calculate the mutual information in the following way:

$$\mathcal{I}(X;Y) = H(X) + H(Y) - H(X;Y)$$

where $H(X)$ is estimated by:

$$\hat{H}(X) = \psi(n) - \psi(k) + \log\left(2^d\right) + \frac{d}{n}\sum_{i=1}^{n}\log\left(\epsilon_Q^k\left(q_i\right)\right)$$

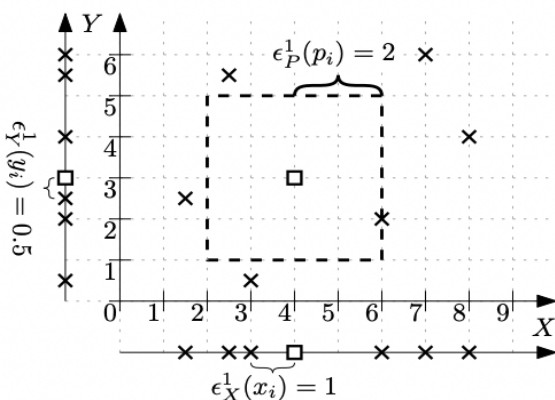

Figure 4: This is the figure from Vollmer et al. (2018) which gives a clear definition of how "3KL" estimate density based on the k-th distance with $L_\infty - norm$

Remarkably, their method was found to consistently estimate entropy, regardless of the specific value chosen for k. This means that as the sample size grows, their approach reliably captures the underlying entropy, showcasing its robustness and applicability.

## A.2 DETAILED ALGORITHM BREAKDOWN FOR MUTUAL INFORMATION INFORMED TWIN PATCH SELECTION.

**Step 1: Initialization.** We slice the first frame, $\mathbf{I}_k$, of a video evenly to $N$ patches, $\{\mathbf{P}_{k,i}\}_{i=0}^{N}$, where the number of patch is a hyper-parameter that could be adjusted so that a patch could best cover an/an complete part of an object. For every patch, we assign a point in the center of the patch, denoted as $\mathbf{p}_{k,i}$, to allow the position of this point to represent the position of the patch, which we call it the "representative point" of a patch.

**Step 2: Tracking to get sequence of 2-D positions.** For all these representative points $\mathbf{p}_{k,i}$ for any given video, we adopt an off-the-shelf tracking model to track those points throughout the video, obtaining a sequence of position in the camera reference frame. For our setting, we choose TAPIR model for tracking. To double check, for a video with length of $F$ frames, we should obtain a

vector of shape $(N, F, 2)$, where N implies there are N representative points and every point has F observation of a 2-D vector representing their $(x, y)$ coordinate in a specific frame.

**Step 3: Incorporating Depth Information to transform to 3-D position sequence** Driven by the principle to best align with real world motion, we consider to turn the observation of 2-D positions given by the tracking model into 3-D position by incorporating depth information. The idea is straightforward: we utilize an off-the-shelf depth estimation model to generate depth for all video frames and then extract the depth value in the position of all representative points for all frames, then we could simply concatenate the depth value into the 2-D position to approximate the real-world 3-D position. After this, the fore-mentioned $(N, F, 2)$ vector becomes of shape $(N, F, 3)$ where the last dimension represent $(x, y, z)$ information. We use MiDaS model to generate depth information.

**Step 4: Mutual information estimation.** For a given video, after we have obtained the $(N, F, 3)$ position vector by going through the above three steps, we could perform mutual information estimation between any two representative points by equation (1). Specifically, for any two representative points, $\mathbf{p}_{k,i}, \mathbf{p}_{k,j}$ in video $\mathbf{S}_k$, we utilize their corresponding $(F, 3)$ position sequence to empirically estimate the mutual information between them.

For a given video $\mathbf{S}_k$, we could obtain a matrix $\boldsymbol{I}_k$ that contains the mutual information between any two patches like below:

$$\boldsymbol{I}_k[i, j] = \widehat{I_{3KL}}(\mathbf{p}_{k,i}, \mathbf{p}_{k,j}), \quad \text{for } i \in \{0, 1, 2, \ldots, N\}, \text{ and } j \in \{0, 1, 2, \ldots, N\} \tag{5}$$

**Step 5: Twin patch determination.** We define the "twin patch" for any given patch $\mathbf{p}_{k,i}$ to be one whose representative point share the highest mutual information with the point of patch of interest. Specifically, the index of twin patch for $\mathbf{p}_{k,i}$, which we denote as $i_{k,i}^{\text{twin}}$ is given as:

$$i_{k,i}^{\text{twin}} = \arg \max_{j \neq k} \boldsymbol{I}_k[i, j], \quad \text{for } i \in \{1, 2, 3, \ldots, N\} \tag{6}$$

By far we illustrate the complete algorithm for determining the twin patch for any given patch in a video. An more concrete way of thinking this is, we hold a dictionary for the whole dataset where the key is any patch and the value is its twin patch who empirically showed to share high mutual information with it.

### A.3 MORE ABLATION STUDIES

**Ablation on number of training samples.** We conducted an ablation study on training datasets of varying sizes to assess the efficacy of our proposed approach. Our investigation encompassed a comparative analysis against seven distinct baseline frameworks. The findings unequivocally demonstrate that regardless of the scale of the training dataset, our method consistently outperforms the corresponding baselines.

Table 4: The ablation study on number of samples for training. We seperately use 1000, 2000 and our complete dataset to perform the pre-training and test the performance on CIDAR-10.

| | N = 1000 | | N = 2000 | | Complete Dataset | |
| Frameworks | Without InfoAug | With InfoAug | Without InfoAug | With InfoAug | Without InfoAug | With InfoAug |
|---|---|---|---|---|---|---|
| SimCLR | 58.97 | **61.30** | 63.97 | **65.37** | 66.64 | **67.48** |
| BYOL | 49.91 | **51.47** | 55.92 | **56.99** | 60.52 | **61.88** |
| SimSiam | 49.94 | **50.13** | 51.86 | **54.14** | 54.22 | **56.69** |
| MoCo | 53.08 | **54.10** | 57.65 | **58.90** | 61.67 | **62.44** |
| NNCLR | 56.33 | **57.62** | 59.59 | **61.54** | 62.97 | **63.57** |
| VICReg | 62.28 | **64.41** | 67.87 | **67.83** | 68.87 | **70.03** |
| TiCo | 50.57 | **55.77** | 58.40 | **61.69** | 62.14 | **63.35** |

Through this evaluation, we explored the impact of dataset size on the performance of our approach. Our findings indicate that the our method transcends the limitations imposed by varying training set sizes. Notably, our results consistently reveal the effectiveness of our approach, showcasing its robustness and generalizability across different data scales. This proves that InfoAug is an independent technique of the data size towards more unified approach of contrastive learning.

**Ablation on numbers of training epoch.** To evaluate the impact of training epochs on model performance, we conducted an ablation study by training our models for different epoch counts:

100, 150, 200, and 250 epochs, respectively. In order to assess the effectiveness of our approach, we compared it against seven different baseline frameworks.

Table 5: The ablation study on training epoch performed on CIFAR-10. We performed on three settings: 150, 200, 250 epochs respectively.

| | 100 Epochs | | 150 Epochs | | 200 Epochs | | 250 Epochs | |
|---|---|---|---|---|---|---|---|---|
| Frameworks | Baseline | InfoAug | Baseline | InfoAug | Baseline | InfoAug | Baseline | InfoAug |
| SimCLR | 62.58 | **63.12** | 67.69 | **70.02** | 66.44 | **67.48** | 67.25 | **68.2** |
| BYOL | 63.75 | **56.39** | 57.81 | **60.36** | 60.52 | **61.88** | 61.55 | **63.77** |
| SimSiam | 54.77 | **57.39** | 55.61 | **57.77** | 54.22 | **56.69** | 56.07 | **59.01** |
| MoCo | 57.55 | **58.02** | 61.27 | 60.73 | 61.67 | **62.24** | 62.33 | **62.34** |
| NNCLR | 59.67 | **61.02** | 62.44 | **65.00** | 62.97 | **63.57** | 63.66 | **65.65** |
| VICReg | 68.45 | **68.52** | 69.74 | **69.83** | 68.87 | **70.03** | 70.08 | 69.09 |
| TiCo | 61.65 | **63.12** | 63.55 | **64.41** | 62.14 | **63.35** | 64.22 | **64.75** |

The results almost consistently demonstrate that our method outperforms the corresponding baselines across various training epochs except 2 entries out of 28 entries. These findings underscore the robustness and efficacy of our approach. It is fair t say that our method showcases superior performance irrespective of the specific number of training epochs employed. This highlights the versatility and generalizability of our approach, suggesting its potential for achieving superior results across diverse training durations and iterations.

**Ablation on 2-layers standard projection head.** In our main comparison experiment and other comparative study, 3-layers projection head was adopted. Now, we also experiment on a standard setting using 2-layers standard projection head as adopted by SimCLR, MOCOv2, BYOL and other SOTA models.

Table 5: The ablation study on a 2-layers projection head on CIFAR10, CIFAR100 and STL10.

| | CIFAR10 | | CIFAR100 | | STL10 | |
|---|---|---|---|---|---|---|
| Frameworks | Baseline | InfoAug | Baseline | InfoAug | Baseline | InfoAug |
| SimCLR | 66.30 | **66.99** | 37.15 | **37.37** | 59.21 | **61.45** |
| BYOL | 59.70 | **61.00** | 29.69 | **32.26** | 54.06 | **55.65** |
| SimSiam | 53.78 | **55.77** | 24.46 | **25.93** | 49.21 | **52.00** |
| MoCo | 60.71 | **61.28** | 30.44 | **32.85** | 55.03 | **55.68** |
| NNCLR | 62.30 | **62.63** | 33.13 | **37.42** | 58.07 | **57.63** |
| VICReg | 70.01 | **70.15** | **41.13** | 40.43 | 62.82 | **61.75** |
| TiCo | 60.91 | **62.73** | 33.23 | **35.03** | 54.16 | **57.91** |

A.4 COMPUTATIONAL OVERHEAD WITH MUTUAL INFORMATION ESTIMATION

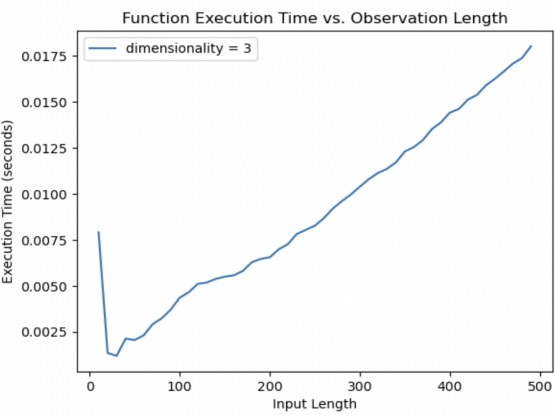

Figure 5: The mutual information empirical estimation time with "3KL" methodology on different numbers of observation

Above gives an diagram on the time cost of mutual information estimation with the "3KL" methodology, with independent variable being the number of observation obtained. The estimation time is measured on a single core of CPU(without any multi-threading). The recent method on mutual information estimation could raise the speed by an order of magnitude and we could also optimized the algorithm by GPU accelerator. For the case of InfoAug, for 50 frames extracted from a video,

we could perform pair-wise mutual information estimation with 40 objects of interest, for less than 2 seconds. It can be concluded that the computational side of InfoAug is rather light-weighted and can be scaled up easily.

