# OpenReview forum: "InfoAug: Mutual Information Informed Augmentation for Representation Learning"
_ICLR.cc/2024/Conference — Submitted to ICLR 2024_

### Official Review · Reviewer_A3xm · 2023-10-25

**Soundness:** 3 good
**Presentation:** 3 good
**Contribution:** 3 good
**Rating:** 6
**Confidence:** 3

**Summary:**

This work proposes a new method for augmenting the pool of positive samples in self-supervised learning (SSL). In particular, the authors make use of videos, that if two patches p1, p2 in the first frame of a video consistently appear together in later frames, they are considered to be a pair of positive samples. The consistency between p1, p2 is measured by the mutual information between their locations in different frame, which itself can be estimated accurately using classic methods. The newly generated positive samples will eventually be used in addition to conventional positive samples in SSL learning. Different projection heads are used for the old and the new positive samples respectively. Experiments on 3 datasets show that the proposed method improves the performance of 7 SSL methods.

**Strengths:**

- The paper is overall well written and easy to follow;
- The method to generate additional positive samples from videos is novel. The idea to treat the whole video as the empirical population of image patches (and to further extract statistical dependence from this population) is also quite interesting;
- The empirical evaluation covers most mainstream SSL methods (up to seven) and three widely-used datasets (CIFAR-10, CIFAR-100, STL-100). Ablation studies regarding the effect of the newly introduced video-based positive samples, the number of training samples and the number of training epochs are conducted and reported carefully;

**Weaknesses:**

- **On twin pairs of video patches** There is, in my opinion, some room for simplification/improvement in the proposed approach. According to the proposed method, patches that are most statistically dependent will be considered to be a pair. Then why not simply take adjacent patches to form a pair, as adjacent patches are more likely to be dependent. Also adjacent patches are visually more consistent/similar. This is conceptually much simpler than your MI-based approach. In this regards, I am also curious about what are the twin patches found in the method. Are they indeed adjacent patches? If not so, an example or a remark will be very helpful and insightful;

- **Applicability of the method**. While the proposed method consistently brings improvement in diverse SSL methods, it relies on high-quality video datasets to work well. As the authors themselves mention, suitable video dataset may not always be available, and in fact this is the reason behind not testing on larger dataset widely used in other SSL literature e.g. ImageNet. This to some extent makes the method not so general as compared to other methods, despite its novel idea to take additional samples from multiple dataset.

- **Complexity**. Due to the use of additional dataset and video processing model (e.g. TAPIR), the overall complexity of the method is also high compared to some other well-known baselines (e.g. SimCLR, Barlow Twins), which is conceptually simpler. It also gives me a sense that the supreme performance of the method is due to the use of external dataset rather than algorithmic/theoretic innovation.

**Questions:**

- See the section `On twin pairs of video patches’ above.
- How do one typically determine the size of the patches in the video framing processing stage? Is it the same as in the main dataset?

---

> ### Author Response · Authors · 2023-11-21
> **Reply to reviewer A3xm**
>
> First of all, thank you very much for your acknowledgement of the article, including the novelty of our ideas and the validity of the method. We also thank you for your time in reviewing the article. In response to your questions, we provide the following answers.
>
>
> 1. First, for the first question, using only neighboring patches as positive samples, although simple in idea, is not rigorous in principle, and will greatly reduce the performance of the model. For example, imagine a dog standing on the edge of a flowerpot, due to their physical proximity, using neighboring patches as positive samples would force the model to push the representation of the dog and the flowerpot closer to each other, however in reality they are not objects of the same class, thus posing significant damage to the model.
>
> 2. For the second question, your suggestion is exactly one of our considerations in this article, we confirmed the effectiveness of InfoAug with great certainty through a large number of comparative and ablation experiments, and we would prefer that subsequent work continues to explore how to manipulate the large in-the-wild dataset to achieve an organic fusion with InfoAug, and to gain the ability to learn from the video dataset with a large amount of unsupervised learning capability from video datasets.
>
> 3. For the third problem, first, we give the following specific data as a reference, for a video data of 80 frames, we only need one second to complete the tracking of nearly 100 points on the whole video. For mutual information estimation, even with the most common algorithm, we only need 0.0025 seconds to complete the mutual information estimation for a pair of random variables with nearly 50 observations. Further, in order to demonstrate the effectiveness of InfoAug, we conducted comparison experiments using randomly selected positive samples, and found that merely increasing the number of iterations (amount of data) does not lead to effective performance improvement, thus illustrating the absolute superiority of mutual information as a selection metric.
>
> 4. Finally, on the question of how to determine the patch size, we are not that clear on what you mean by "main dataset". In any case, patch size is an hyperparameter that should be determined by roughly estimating the size of individual objects contained in the video dataset. Although we found in our experiments that the choice of this hyperparameter does not bring much effect variation, it is still a case-by-case (dataset) choice, and the ultimate goal is to expect the patch size to match the majority of the objects in the scene to get the best possible cover effect.
>
>
> Finally, thank you for your questions, especially the first one which is intriguing. We hope our answers have solved your confusion.

---

> > ### Comment · Reviewer_A3xm · 2023-11-22
> > **Reply to authors**
> >
> > Many thanks for your detailed rebuttal, which addresses most of my confusion. I'll keep my already positive score.

---

### Official Review · Reviewer_xZ59 · 2023-11-01

**Soundness:** 2 fair
**Presentation:** 2 fair
**Contribution:** 2 fair
**Rating:** 3
**Confidence:** 5

**Summary:**

The paper introduces InfoAug, a novel data augmentation technique that identifies positive pairs of patches in a video frame by estimating their mutual information using off-the-shelf tracking models. A dual branch is utilized to handle the mutual-information-guided positive pairs.

**Strengths:**

- The concept of identifying positive patches is interesting.
- Using estimated mutual information to find positive patches seems reasonable.
- Experiments show the effectiveness of the positive patches selection and the dual branch.

**Weaknesses:**

- The experiments are primarily conducted on small datasets and employ small backbones like ResNet-18, which may limit the generalizability of the results.
- Fairness of Comparisons: The comparison with other methods could be improved. Specifically, the use of a 3-layer head instead of the standard single-layer might lead to skewed results. Additionally, the direct comparison of InfoAug with other methods is not fair due to the employment of 2 branches in InfoAug. The improvements over the "random twin patch" are marginal and sometimes worse, as seen in Tables 1 and 2.
- Training Epochs: The models are trained for only 100 epochs, which might not be sufficient for convergence. When more epochs are used, as shown in Table 5, the improvements seem to diminish. Again, it is not fair to compare them directly.

**Questions:**

- Patch Extraction: Could the authors clarify how patches are extracted? Are they overlapping or non-overlapping? Are they selected from specific regions or chosen randomly?
- Application to Image Datasets: How does InfoAug select positive pairs from video datasets and apply this knowledge to image datasets? Is the model first pre-trained on video datasets and then fine-tuned on image datasets, or are twin patches retrieved from video datasets during training on image datasets?

---

> ### Author Response · Authors · 2023-11-22
> **Reply to reviewer xZ59**
>
> First of all, we sincerely appreciate the valuable time of the reviewer and thank you for your questions and corrections. We also appreciate your recognition of the novelty and the acknowledgment of the use of mutual information and dual branch formulation in our article.
>
> Now, I will respond to your corrections and questions in a sequential order:
>
> 1. Regarding the first concern, we appreciate that you noticed our argument about the size of the pretraining dataset. Since the existing InfoAug is well-suited for learning from small to medium-sized datasets by imitating the learning process of humans through mutual information estimation, we considered that using a larger model such as ResNet-50 for training would lead to overfitting, as ResNet-18 is already a large model for the existing dataset scale. This way, we could not effectively showcase the performance of InfoAug itself.
>
> 2. For the second concern, we agree with your suggestion, and thus, we examined prominent state-of-the-art models such as SimCLR, BYOL, MoCoV2, which indeed employ a two-layer fully connected head as the projection head. Consequently, we conducted experiments based on this suggestion, and the results are put into Appendix A.3. The results still, under standard 2-layer projection head, show a consistent improvement over all the SOTA models. As for your concern about "random-twin-patch," I believe there might be a misunderstanding regarding the purpose of that experiment. We intentionally used randomly selected patches to illustrate the importance of mutual information as the metric for selection. Randomly selecting patches as positive samples without considering mutual information would not yield good results. This experiment was intentionally done to provide a comprehensive demonstration of the significance of mutual information in InfoAug, as a comparative study to only adopt random selection.
>
> 3. Regarding the third concern, we chose 100 epochs as the main experiment, as we found that it almost reached convergence within that timeframe. The reason behind this is that we used patches as training units instead of whole images, which increased the number of iterations per epoch by a factor of N (where N is the number of patches in an image).
>
>
> Regarding your questions:
>
> 1. Certainly, we would be pleased to explain the specific process of patch selection. Each image is divided into N non-overlapping patches. Although using overlapping patches may further improve performance, we intentionally avoided it to demonstrate the true effectiveness of InfoAug. Overlapping patches may contain the same object, which would not clearly illustrate the necessity of mutual information guidance. Additionally, the patches are uniformly divided without any artificial design, intended to not hurt its generalizability.
>
> 2. For the second question, yes, the model is first pretrained on a video dataset without labels and then fine-tuned and tested on an image dataset. This aligns with the designed workflow of InfoAug and resembles the way humans learn.
>
> We hope our additional experiment and explanation shall address your concern, and could be seen by more, allowing for further consideration of integrating mutual information into contrastive learning.

---

### Official Review · Reviewer_m3jm · 2023-11-01

**Soundness:** 2 fair
**Presentation:** 2 fair
**Contribution:** 1 poor
**Rating:** 3
**Confidence:** 3

**Summary:**

The rough idea of the proposed method is to construct a video-based dataset that can be applied to unsupervised image representation learning. In specific, for a given patch, mutual information is estimated between patches via applying a pre-trained tracking module. Then, for every patch, the similar patch with the highest mutual information is selected as a twin patch. Then, the assigned twin patch is regarded as a pseudo-positive pair on the contrastive learning scheme. Authors experiment the efficacy of proposed method on the CIFAR-10/100 and STL dataset with integration to existing self-supervised learning algorithm.

**Strengths:**

1. The idea of extracting patch-wise information in the video dataset and applying to benefit 2d dataset training is intruiging.
2. The proposed loss function is simple

**Weaknesses:**

1) My biggest issue in this paper is the significance of the result. The linear probing result on CIFAR-10/100 is strictly underwhelming, given that conventional self-supervised learning result on the given dataset approaches around 92.6% in CIFAR-10 and 70.5% in CIFAR-100 (see [1]). In contrast, BYOL in the author's code shows 60.5% in CIFAR-10 and 30.3% in CIFAR-100.\
2) Due to this result, I am puzzled as to why we should look at this "pre-training on the video dataset". The method seems like an underwhelming training strategy albeit using additional data.\
3) Thereby, I suggest the authors redesign the experiment by pre-training on a much larger dataset and show better performance on such CIFAR-10/100.\
4) The paper has some typos (utual information, (z1,z1) before equation (4), etc...).\
5) Furthermore, consider incorporating other models (e.g. CLIP) that can be applied in a zero-shot manner.

In summary, I am severely concerned about the validity of the proposed pre-training approach since the result in Table 1 is very underwhelming and far from the recent numbers. Thereby, I lean to rejection.



***References***
[1] Unsupervised Visual Representation Learning via Mutual Information Regularized Assignment, NeurIPS 2022

**Questions:**

See Weakness

---

> ### Author Response · Authors · 2023-11-22
> **Reply to reviewer m3jm**
>
> First of all, thank you very much for your time devoted in giving us the valuable feedback and your acknowledgement on the novelty of our method and design of loss function.
>
> With respect to the questions you raised, we would like to provide the following responses:
>
> 1. Regarding the doubts about the first, second, and third points, we offer the following explanation: We aim to provide a more rigorous method for defining positive samples, inspired by human learning. In a given environment, we observe various objects and obtain a rough estimate of the mutual information between objects. One implicit requirement of this method is that we have a sufficient number of observations of objects in an environment. This requirement is naturally fulfilled in human learning, but it is severely violated in large-scale in-the-wild video datasets. These datasets often suffer from significant camera movements and occasional scene changes. Sometimes, an object only appeared for first frame and quickly left the camera in subsequent frames. This setup is not consistent with our human learning perception and does not clearly demonstrate the intended concept of InfoAug, which aims to judge positive and negative samples based on the moderate perception of information sharing between objects. Although we have conducted corresponding experiments on large-scale datasets, even achieving results comparable to traditional SOTA models, we did not focus solely on this aspect. It is unrelated to the experimental results but only mismatched with the method we wanted to convey. Therefore, we explicitly explained this mismatch with large in-the-wild datasets in our paper, hoping that future work can further expand this method and find unrestricted learning approaches for InfoAug on large-scale datasets. Nevertheless, we believe this should not hinder the demonstration of InfoAug's powerful features. Through a combination of 21 models and datasets, 3 comparative experiments, and ablative experiments across 5 scenarios, we demonstrated the improvement of InfoAug over the existing SOTA baseline, proving the interesting connection between mutual information and positive/negative samples. We hope this method of defining positive and negative samples can be further popularised.
>
> 2. Regarding your fourth point, we sincerely appreciate you pointing out the issue, and we have corrected the errors in the paper. To present more clearly the improvement of InfoAug over the existing SOTA and provide more detailed usage instructions, we have supplemented the appendix with experiments and calculations of the computational cost of mutual information on different projection heads. Additionally, we have made modifications to two figures in the paper to enhance readers' understanding of the natural workings of InfoAug. As for your fifth suggestion, we greatly appreciate your proposal and assistance. However, since CLIP is a multimodal model for language and image embedding, and our method does not involve the language modality, we would appreciate it if you could provide further clarification and explanation.
>
> Thanks!

---

> ### Comment · Reviewer_m3jm · 2023-11-23
>
> Thank you for your response.
> However, I think the author's response does not clarify any of my concerns, especially on the comparison against existing methods. It is hard to understand the rebuttal despite the simple question of why the baseline method's performance does not match the reported numbers (I believe reviewer LYQ4 asks similar questions). I'll stick to my current score.

---

### Official Review · Reviewer_LYQ4 · 2023-11-04

**Soundness:** 1 poor
**Presentation:** 2 fair
**Contribution:** 1 poor
**Rating:** 3
**Confidence:** 4

**Summary:**

This paper proposes InfoAug, a contrastive learning framework that employs a novel way of selecting positive pairs. Besides positive pairs that are generated by augmentations, InfoAug also selects the "twin patch" which maximizes the mutual information of the original patch as another positive patch. Experiments show that when trained on a video dataset and evaluated on CIFAR-10, STL-10, and CIFAR-100, the proposed InfoAug brings 1-2% improvement over existing contrastive learning methods.

**Strengths:**

This paper proposes a novel way of selecting positive pairs in contrastive learning, which is to select a twin patch that maximizes mutual information between the two pairs. Such a design proposes a new direction in utilizing video datasets in contrastive learning.

**Weaknesses:**

Overall, the major problem of this paper is the weak experimental results. The experiment results are weak in two aspects:
1. The accuracy of baselines on evaluated datasets is low. For example, when trained and tested on CIFAR-10, SimCLR can achieve >90% accuracy. However, in this paper, when trained on DAVIS20+GMOT40 and tested on CIFAR-10, the performance is only 60%-70%. I would suggest including CIFAR-10 as a pseudo-video dataset in the pre-training stage so that the paper can make a fair comparison with current methods on these datasets.
2. The improvement of the proposed method is marginal. For example, when the accuracy on CIFAR-10 is 60%-70%, the standard deviation can be large, but the proposed method only improves the performance by 1-2% on each dataset. This makes the improvement shown in the paper not convincing enough.
3. The paper does not include an analysis of computation overhead over baseline methods. Estimating the mutual information between two patches can introduce some computation overheads, making the framework slower than baseline methods.

The presentation of the paper is also not very clear and needs further polishing. There are multiple typos and grammar errors, and the font in the figures is too small.

**Questions:**

See weaknesses

---

> ### Author Response · Authors · 2023-11-22
> **Reply to reviewer LYQ4**
>
> Thank you very much for your time in reviewing our work, giving valuable feedback. We also thank you for giving credit that it serves as novel for contrastive learning in video dataset.
>
> Now, we will address your concern down below in a sequential order:
>
> 1. Regarding your concern 1 and 2, yes, we use a small-scale dataset for pretraining, which leads to a downscaled accuracy on the downstream task. The reason behind that lies in what we would like to contribute with this paper: To demonstrate and elicit a natural definition of “positive samples” in contrastive learning. The inspiration lies in how we human recognizes positive samples: we learn in an environment with necessary amount of observation (you cannot see two cats walking together for one millisecond and learning they are of same species). The problem with current large scale video dataset is that, the scene is dynamically moving, leading to extremely insufficient observation for almost any object, which greatly differs from human-learning settings. So, although some of the result looks competitive with the traditional contrastive learning paradigm on those in-the-wild datasets, which exhibits high camera dynamic, the result is a compromised one in that the advantage of InfoAug is offset by the divergence of the intentional learning setting and that of large video dataset shoot casually, which might be misleading to the question of “what is the strength and how to apply it”. Therefore, we make clear our intention and proposed learning setting explicitly in the paper to avoid the misunderstanding of the suitable application of InfoAug, and test its performance on video dataset that do offers similar level of observation for a group of objects. The main results on three major downstream dataset, ablation study on loss weight and training epoch, all consistently shows the benefit of InfoAug as a valuable approach for defining positive samples. We notice the fact that the improvement ranges from 1% to 4% and we hope the improvement will be larger if the scale can be further expanded by future work. To this extent, we also examined if InfoAug really bring significant result by doing various comparison experiment to ablate out all the possible contributing factor, including the increased training sample, dual-branch formulation. and it shows that all the consistent contribution vanish, without using InfoAug to provide mutual-information awareness, demonstrating the absolute value in defining “positive sample” by InfoAug.
>
> 2. For your third concern, we attach the relevant diagram at the end of appendix for your valid concern. For a robot to determine the positive samples in a video, where we assume it focus simultaneously on as much as 40 objects, for an interpolation of 50 frames, it only took 2 seconds on a single CPU core. If, as we did by multi-thread it with 16 working thread, it is reduced to less that 0.2 second. For the most updated MI estimation technique, it could be further reduced to 0.1 second without any multi-threading, without GPU-acceleration.
>
> In summary, we show by multiple comparative experiment (please see more in section 4.1, 4.2 and 4.3)that the improvement is significant and consistent over any SOTA model-dataset combination, that the way of defining positive samples by mutual information is both natural and of great potential, we hope such an idea could be seen by others to together expand it towards to more unified contrastive learning approach.

---

### Meta-Review · Area_Chair_HR63 · 2023-12-05

**Metareview:**

Three reviewers have serious concerns about the experimental results which did not validate the effectiveness of the proposed method.

**Justification For Why Not Higher Score:**

N/A

**Justification For Why Not Lower Score:**

N/A

---

### Decision · Program_Chairs · 2024-01-16

Reject